# Co-Delivery of an Innovative Organoselenium Compound and Paclitaxel by pH-Responsive PCL Nanoparticles to Synergistically Overcome Multidrug Resistance in Cancer

**DOI:** 10.3390/pharmaceutics16050590

**Published:** 2024-04-26

**Authors:** Daniela Mathes, Letícia Bueno Macedo, Taís Baldissera Pieta, Bianca Costa Maia, Oscar Endrigo Dorneles Rodrigues, Julliano Guerin Leal, Marcelo Wendt, Clarice Madalena Bueno Rolim, Montserrat Mitjans, Daniele Rubert Nogueira-Librelotto

**Affiliations:** 1Programa de Pós-Graduação em Ciências Farmacêuticas, Universidade Federal de Santa Maria, Av. Roraima 1000, Santa Maria 97105-900, Brazil; danielamathes1609@gmail.com (D.M.); leticiabuenomacedo@gmail.com (L.B.M.); biancamaia662011@gmail.com (B.C.M.); clarice.rolim@gmail.com (C.M.B.R.); 2Laboratório de Testes e Ensaios Farmacêuticos In Vitro, Departamento de Farmácia Industrial, Universidade Federal de Santa Maria, Av. Roraima 1000, Santa Maria 97105-900, Brazil; tais.pieta@acad.ufsm.br; 3Laboratório de Engenharia e Processos Químicos, Universidade Federal de Santa Maria, Av. Roraima 1000, Santa Maria 97105-900, Brazil; 4Departamento de Química, Universidade Federal de Santa Maria, Av. Roraima 1000, Santa Maria 97105-900, Brazil; rodriguesoed@gmail.com (O.E.D.R.); jleal@iffarroupilha.edu.br (J.G.L.); marcelowendt@hotmail.com (M.W.); 5Departament de Bioquimica i Fisiologia, Facultat de Farmacia i Ciències de l’Alimentaciò, Universitat de Barcelona, Av. Joan XXIII 27-31, 08028 Barcelona, Spain; 6Institute of Nanoscience and Nanotechnology, Universitat de Barcelona, 08028 Barcelona, Spain

**Keywords:** MDR tumor cells, in vitro assays, 3D tumor model, spheroids, selenium compound

## Abstract

In this study, we designed the association of the organoselenium compound 5′-Seleno-(phenyl)-3′-(ferulic-amido)-thymidine (AFAT-Se), a promising innovative nucleoside analogue, with the antitumor drug paclitaxel, in poly(ε-caprolactone) (PCL)-based nanoparticles (NPs). The nanoprecipitation method was used, adding the lysine-based surfactant, 77KS, as a pH-responsive adjuvant. The physicochemical properties presented by the proposed NPs were consistent with expectations. The co-nanoencapsulation of the bioactive compounds maintained the antioxidant activity of the association and evidenced greater antiproliferative activity in the resistant/MDR tumor cell line NCI/ADR-RES, both in the monolayer/two-dimensional (2D) and in the spheroid/three-dimensional (3D) assays. Hemocompatibility studies indicated the safety of the nanoformulation, corroborating the ability to spare non-tumor 3T3 cells and human mononuclear cells of peripheral blood (PBMCs) from cytotoxic effects, indicating its selectivity for the cancerous cells. Furthermore, the synergistic antiproliferative effect was found for both the association of free compounds and the co-encapsulated formulation. These findings highlight the antitumor potential of combining these bioactives, and the proposed nanoformulation as a potentially safe and effective strategy to overcome multidrug resistance in cancer therapy.

## 1. Introduction

Cancer is characterized by a set of diseases with sustained proliferation signaling, which causes excessive cell division cycles with phenotypic and molecular changes [1,2]. Treatments for cancer are mainly limited to surgery, radiotherapy, immunotherapy, and chemotherapy, which are still closely associated with non-specific toxicity and problems of multidrug resistance (MDR) [3].

Nucleoside analogues are important tools that can be used in anticancer treatments. Identified as antimetabolites, and being similar to intrinsic nucleosides, they are phosphorylated and converted to analogous nucleotides, and can then have both the ability to inhibit DNA polymerase or ribonucleotide reductase and the ability to insert themselves into DNA and inhibit its synthesis [4]. However, resistance to nucleoside analogues is predominant, resulting from difficult conversion to their active metabolites or restricted absorption by tumor cells, often attributed to decreased expression of nucleoside transport proteins [4,5].

Paclitaxel (PTX) is an important antitumor drug classified as antimitotic and antimicrotubule. Its action is due to the stabilization of microtubules, induction of mitotic arrest, and consequent apoptosis of cells. It can be used to treat advanced prostate and breast carcinomas, lung carcinoma, endometrial cancer, bladder cancer, cervical carcinoma, and some others. However, several problems have been associated with this medication, such as hair loss, allergic reactions, nausea, neutropenia, leukopenia, anemia, arthralgia, myalgia, mucositis, weakness, and neuropathy. Another important point is the associated low bioavailability, of around 10–30% [6,7].

Strategies for improving bioavailability and efficacy, and reducing nonspecific toxicity, can be used in the treatment of cancer, including combined therapies and the advantages of nanotechnology-based drug delivery systems. The heterogenicity of cancer can be more easily treated when considering the fusion of anticancer compounds in the same therapy, that is, a combined therapy. There are several advantages associated with it, which include reducing the effect of drug resistance, reducing metastatic potential, and reducing therapeutic doses that would have to be used individually [8]. If the interaction enhances treatment efficacy and has a collective effect greater than the sum of individual medications, then the combination is deemed synergistic [9].

Polymeric nanoparticle systems allow various adaptations and direction of formulations. Passive targeting of enhanced permeability and retention (EPR) is one of the important possibilities presented by nanoformulations, as the tortuous and immature vasculature of tumors allows small particles to accumulate at the site. Furthermore, targeted delivery helps reduce toxicity in normal cells, can help protect drugs from degradation, and increase half-life, loading capacity, and solubility [10]. In the same sense, nano-scale pH-sensitive systems offer a viable approach for enhancing targeted drug delivery to tumors, since the pH disparities between cancerous tissues (ranging from approximately pH 6.6 to 5.4) and healthy tissues (with a pH of 7.4) are readily discernible [11].

Facing the limitations of individual treatments and the potentialities of the nanotechnology, here we proposed a combined therapy of PTX and a novel organoselenium nucleoside analogue, 5′-Seleno-(phenyl)-3′-(ferulic-amido)-thymidine (AFAT-Se), which has shown promising antitumor effects in a previous in vitro study [12]. The rationale of the combination of these bioactive compounds relies on the potential to achieve an effective treatment that could synergically overcome MDR in cancer cells. Using different drugs that act by different mechanisms of action is likely to achieve greater antineoplastic activity and sensitization of resistant tumor cells, and a decrease in side effects due to the reduction in therapeutic doses of each drug. Combined with these advantages, the co-encapsulation of the bioactive compounds in a nano-based system is expected to result in further improvements in the therapeutic outcomes.

In this study, an effective synergistic combination of paclitaxel with a novel organoselenium nucleoside analogue, AFAT-Se, was found. Furthermore, these bioactive compounds were effectively and satisfactorily co-loaded into a nanoformulation containing a pH-dependent surfactant (77KS), a stabilizing surfactant capable of increasing tumor sensitization (poloxamer 407), and the polymer poly(ε-caprolactone) (PCL). The role of the pH on the membrane-lytic activity of the pH-sensitive NPs was evaluated using the erythrocyte as a model for the endosomal membrane. Furthermore, the safety of NPs was evaluated by hemocompatibility assay, and their nonspecific cytotoxicity was evaluated using two non-tumor cell models. The antitumor activity, synergistic effects, and potential for overcoming the MDR effect were evaluated using an NCI/ADR-RES-resistant cell line by means of two-dimensional (2D) and three-dimensional (3D) in vitro platforms.

## 2. Materials and Methods

### 2.1. Materials

HPLC-grade acetonitrile was acquired from Tedia (Fairfield, CT, USA). The polymer poly(ε-caprolactone), with a molecular weight within the range of 70,000 to 90,000, as well as Pluronic F-127 (poloxamer 407), 2,5-diphenyl-3-(4,5-dimethyl-2-thiazolyl) tetrazolium bromide (MTT), dimethyl sulfoxide (DMSO), phosphate-buffered saline (PBS), Dulbecco’s Modified Eagle’s Medium (DMEM) supplemented with L-glutamine (584 mg/L), fetal bovine serum (FBS), antibiotic solution (with 10,000 units penicillin and 10 mg streptomycin/mL), and trypsin-EDTA solution (comprising 0.5 g of porcine trypsin and 0.2 g of EDTA·4Na per liter of Hank’s Balanced Salt Solution), were sourced from Sigma-Aldrich (São Paulo, SP, Brazil). Paclitaxel (PTX) was purchased from Zibo Ocean International Trade (Zibo, Shangdong, China).

The organoselenium compound AFAT-Se was obtained from the LabSelen-NanoBio (Federal University of Santa Maria, Santa Maria, Brazil). This organoselenium was synthesized and fully characterized as previously described [12].

### 2.2. Analytical Method

The purity verification, quantification, and stability study of the compounds under analysis were carried out by reversed-phase liquid chromatography (RP-LC). The chromatography method for the simultaneous analysis of AFAT-Se and PTX was optimized on a Shimadzu LC system (Shimadzu, Kyoto, Japan) containing an SPD-M20A photodiode array (PDA) detector, using a Gemini NX C18 Phenomenex column (150 mm × 4.6 mm; 5 μm). The system was operated in isocratic mode with flow gradient (0.8–1.2 mL/min), room temperature, mobile phase containing ultrapure water, acetonitrile, and methanol (37:38:25, *v*/*v*), and the UV detection was set at 227 nm.

### 2.3. pH-Responsive Surfactant Used in Nanoparticles

In the composition of the nanoparticle formulations, a distinct anionic surfactant derived from the amino acid lysine, 77KS, was included [13,14]. This surfactant originates from N^α^,N^ε^-dioctanoyl lysine, complemented by an inorganic sodium counterion having a molecular weight of 421.5 g/mol and critical micelle concentration (CMC) of 3 × 10^3^ g/mL. The synthesis of this surfactant has been previously described [13].

### 2.4. Preparation of Nanoparticles

The nanoparticles containing AFAT-Se and the antitumor paclitaxel were prepared by nanoprecipitation, a method developed by Fessi and co-workers (1988) [15]. Firstly, an organic phase containing 19 mg sorbitan monooleate (Span 80^®^, Sigma-Aldrich, São Paulo, SP, Brazil) and 25 mg PCL was prepared in 10 mL of acetone and maintained for 60 min under magnetic stirring at 45 °C. Then, 10 mg of AFAT-Se, previously dissolved in 3 mL of MeOH and 2 mg of paclitaxel (PTX) dissolved in 5 mL of acetone, were added to this organic solution (ratio 5:1 of AFAT-Se:PTX). At the same time, an aqueous dispersion (20 mL) containing poloxamer 407 (75 mg) was obtained and the pH-sensitive surfactant 77KS (2.5 mg) was also added to the aqueous solution. Then, the organic phase was poured into the aqueous phase under magnetic stirring and maintained for 20 min. Finally, the organic solvent was eliminated by evaporation under reduced pressure until reaching 5 mL of final volume (AFAT-Se-PTX-PCL-77KS-NP suspension).

NPs without both the bioactives (PCL-77KS-NP), and without the pH-sensitive adjuvant 77KS (AFAT-Se-PTX-PCL-NP and PCL-NP) were also prepared when necessary for tests that required such formulations. Furthermore, a solution containing the association of free oganoselenium and paclitaxel was also used for comparative purposes (AFAT-Se-PTX-Free). Finally, solutions separately containing the free form of each bioactive were also prepared (PTX-Free and AFAT-Se-Free), as well as the corresponding NPs without the co-loading (PTX-PCL-77KS-NP and AFAT-Se-PCL-77KS-NP).

### 2.5. Characterization of Nanoparticles

The hydrodynamic diameter and polydispersity index (PDI) of nanoparticles (NPs) were determined using dynamic light scattering (DLS) using a Malvern Zetasizer ZS (Malvern Instruments, Malvern, UK), with a sample size of n = 3. Dilution of all samples in ultrapure water at a ratio of 1:500 (*v*/*v*) preceded each measurement, involving a minimum of three series, each with ten consecutive runs. Zeta potential (ZP) analyses were performed using the same equipment, evaluating the electrophoretic mobility of the formulations in 10 mM NaCl aqueous solution (1:500 *v*/*v*) in at least three sets, each comprising ten sequential runs. Furthermore, pH measurements were obtained directly from NP suspensions at room temperature using a calibrated potentiometer (UB-10; Denver Instrument, Bohemia, NY, USA).

The morphology of the co-loaded NP (AFAT-Se-PTX-PCL-77KS-NP) was evaluated employing scanning electron microscopy (SEM) (JEOLJSM 6360, Akishima, Japan). In this process, 20 µL of the NP suspension was deposited onto a stub and incubated for 12 h at ambient temperature. Subsequently, the stub was subjected to gold coating under diminished pressure, followed by examination of the samples utilizing a voltage of 10 kV.

### 2.6. Entrapment Efficiency

Quantification of drug entrapment efficiency (EE%) was carried out using a methodology consisting of Amicon Ultra-0.5 centrifugal filters with a molecular weight cutoff of 10,000 Da (MWCO, Millipore, Cork, Ireland). A determined volume of nanoparticles (NPs) was submitted to the device and taken to a centrifuge for 20 min at 3610× *g*, thus isolating the free compounds, and then quantified using the reversed-phase liquid chromatography technique already described. The following equation was used for the calculations.
$$EE\%=\left[\frac{\left(Total\;content-Free\;content\right)}{total\;content}\right]\times 100\%$$

### 2.7. In Vitro Antioxidant Activity

DPPH and ABTS assays were used to determine scavenging activity [16,17]. Co-loaded NP and associated free AFAT-se and PTX were evaluated. Increasing concentrations in μg/mL of the association of free and co-encapsulated NP were used in a ratio of 5:1 (AFAT-Se:PTX). Therefore, the concentrations were 0.1:0.02, 1.0:0.2, 5.0:1.0, 10:2.0, and 25:5.0 μg/mL. The two protocols are similar but complementary. In the case of the DPPH assay, the samples (75 µL) were dispensed into 96-well plates containing a 50 mM 2,2-diphenyl-1-picrylhydrazyl (DDPH) solution (150 µL) in methanol, followed by incubation in darkness at room temperature for 30 min. After that, the absorbance of the samples was measured at 550 nm using a Multiskan FC microplate reader (Thermo Fisher Scientific, Shanghai, China). In the same sense, 2,2′-azino-bis(3-ethylbenzothiazoline-6-sulfonic acid) (ABTS) has a similar protocol. However, the initial solution required prior preparation, in which 5 mL of 7 mM ABTS in water was mixed with 88 µL of 140 mM sodium persulfate, and kept in the dark at room temperature for 12 h. To obtain the final solution, the previous one was diluted in 10 mM phosphate solution pH 7.0 to obtain 42.7 µM of ABTS; then, the same protocol described for DPPH was followed, but the absorbance was measured at 734 nm. To determine potential turbidity interference from nanoparticles, blank absorbance was determined by preparing solutions containing the increasing concentrations of NPs and 150 µL of methanol or water, substituting DPPH or ABTS, respectively. Negative control was determined associating the DPPH or ABTS solutions with 75 µL of water. The following equation was used for the calculations.
$$\%Scavenging\;actitvity=\left[\frac{\left(Sample-Blank\right)\times 100}{Negative\;control}\right]-100$$

### 2.8. In Vitro Cell Biological Safety Profile

To evaluate the safety of the proposed formulation, two approaches were used. Non-specific cytotoxicity was evaluated on non-tumor cell line 3T3 (murine Swiss albino fibroblasts), and also, of equal importance, on human mononuclear cells of peripheral blood (PBMCs). In the first method mentioned, cell cultures were maintained in Dulbecco’s Modified Eagle Medium (DMEM) with 4.5 g/L of glucose, supplemented with 10% (*v*/*v*) FBS, and incubated at 37 °C with 5% CO_2_. At 80% confluency, cells were seeded at a density of 6.5 × 10^4^ cells/mL in 96-well cell culture plates and incubated for 24 h. Then, the cells were treated and maintained for 24 h at 37 °C with 5% CO_2_, and finally the cell viability was determined using the MTT assay. The methodology involving PBMCs establishes their isolation from human blood, in accordance with the guidelines established by the Research Ethics Committee of the Federal University of Santa Maria, Brazil (CAAE protocol 44017921.3.0000.5346). PBMCs were extracted using a density gradient centrifugation technique with Histopaque^®^-1077 (Sigma–Aldrich, São Paulo, Brazil). Thus, they were isolated and then cultivated in RPMI-1640 medium, fortified with 10% (*v*/*v*) FBS. Thereafter, these cells were dispensed into a 96-well plate in order to guarantee a cell density of 5 × 10^5^ cells/mL, treated for 24 h, and incubated with MTT (0.5 mg/mL) for 3 h at 37 °C with 5% CO_2_. The absorbances were measured at 550 nm using a Multiskan FC microplate reader (Thermo Fisher Scientific, Shanghai, China).

It is worth mentioning that, in both methodologies, AFAT-Se-PTX-PCL-77KS-NP, PCL-77KS-NP, and AFAT-Se-PTX-Free were evaluated. Likewise, the concentrations used were in μg/mL in an increasing proportion of 5:1 (AFAT-Se:PTX). Therefore, the concentrations used were 1.0:0.2, 3.0:0.6, 5.0:1.0, 15:3.0, and 30:6.0 μg/mL.

### 2.9. Hemocompatibility Studies

The analysis for blood compatibility was carried out using a hemolysis assay [18,19]. For this, erythrocytes from healthy volunteers were used in accordance with the guidelines established by the Ethics Committee in Research, from the Federal University of Santa Maria, Brazil (protocol CAAE 44017921.3.0000.5346). After isolating the erythrocytes through centrifugation, they were washed and suspended in isotonic PBS pH 7.4; 300 mOsmoL/L at a cell density of 8 × 10^9^ cells/mL. For this test, the formulations AFAT-Se-PTX-PCL-77KS-NP, PCL-77KS-NP, and AFAT-Se-PTX-Free were used, and the concentrations in this assay were in μg/mL in an increasing proportion of 5:1 (AFAT-Se:PTX). Therefore, the concentrations used were 240:48, 320:64, and 400:80. This was followed by an incubation period of 5 h with 25 µL of the erythrocyte suspension under slow agitation; the reaction was stopped by centrifugation at 10,000 rpm for 5 min. Positive and negative controls were prepared by dispersing 25 µL of the erythrocyte suspension in water and PBS, respectively. Subsequently, a 200 µL aliquot of each sample was dispensed into 96-well plates, and the absorbance measurement was read at 550 nm employing a microplate reader (Multiskan FC, Thermo Fisher Scientific, Shanghai, China).

### 2.10. pH-Dependent Membrane-Lytic Activity of Nanoparticles

To evaluate the membrane-lytic capacity of the NPs according to pH, a methodology similar to the hemocompatibility assay was applied. However, in this experimental procedure, the erythrocytes were used as endosomal membrane models [5]. The NP suspension was diluted in PBS pH 7.4, 6.6, or 5.4 at the concentrations of 240:48, 320:64, and 400:80 µg/mL of AFAT-Se:PTX and kept in contact with 25 μL of the erythrocyte suspension under gentle agitation for 5 h at room temperature. Positive and negative controls were prepared by dispersing 25 µL of the erythrocyte suspension in water and PBS, respectively. To stop the reaction, the samples were centrifuged at 10,000 rpm for 5 min. Finally, readings were performed in a 96-well plate at 550 nm employing a microplate reader (Multiskan FC, Thermo Fisher Scientific, Shanghai, China).

### 2.11. Synergic in Vitro Antitumor Activity Using 2D Cell Model

The multidrug-resistant (MDR) cell line NCI/ADR-RES (human ovarian cancer cells) was used for the assay. This cell line was kindly donated by Dr. Antoni Benito from the University of Girona (Spain), and cultured continuously in DMEM medium (4.5 g/L glucose) containing 1.0 µg/mL of doxorubicin. The cells were grown in 75 cm^2^ growth flasks, under precise conditions at 37 °C, within a controlled atmosphere containing 5% CO_2_. This cultivation process was sustained until the cells reached an approximate confluence of 80%. Then, the cells were seeded in 96-well plates and subjected to incubation at 37 °C in a controlled and humidified environment enriched with 5% CO_2_ for a period of 24 h. Then, the cells were treated with the following formulations: AFAT-Se-PTX-PCL-77KS-NP, AFAT-Se-PTX-Free, PTX-PCL-77KS-NP, PTX-Free, AFAT-Se-PCL-77KS-NP, AFAT-Se-Free, and PCL-77KS-NP. In this way, it was possible to evaluate the activity of the separated free bioagents and the synergism of the association, as well as the presence of the nanoformulation in this same environment, and also the formulation without the active compounds. The concentration of AFAT-Se regardless of the formulation was always 1, 3, 5, 15, and 30 µg/mL. The PTX concentration was 0.2, 0.6, 1, 3, and 6 μg/mL. The cells were treated for 72 h and then incubated for 3 h at 37 °C, 5% CO_2_, with 0.5 mg/mL of the MTT solution in DMEM without FBS. Subsequently, the medium was replaced by 100 µL of dimethyl sulfoxide (DMSO), and the absorbance was quantified at 550 nm utilizing a Microplate Reader (Multiskan FC, Thermo Fisher Scientific, Shanghai, China). Results are expressed as percentage of viability relative to untreated control cells. The acquired results were utilized to calculate the combination index (CI) using the median-effect method with CompuSyn software (ComboSyn, Inc., Paramus, NJ, USA, version 1.0). In accordance with the Chou–Talalay method, synergism, an additive effect, and antagonism, are indicated by CI values below 0.9, between 0.9 and 1.1, and above 1.1, respectively [20,21,22].

### 2.12. In Vitro Antitumor Activity Using 3D Cell Models

The hanging drop method was used to obtain spheroids from the NCI/ADR-RES cell line [23,24,25]. After the growth process, small aliquots of 20 µL of cell suspension at a concentration of 3 × 10^5^ cells/mL were carefully dispensed into the lid of a Petri dish. Following this step, the lids, now inverted, were positioned above dishes containing 10 mL of sterile ultrapure water. These were then incubated in a controlled environment at 37 °C with a humidified atmosphere with 5% CO_2_, initiating conditions conducive to the formation of cell aggregates. Each aggregate formed was transferred to a well of a 96-well plate pre-coated with agarose. The medium used to maintain the aggregates was DMEM 10% FBS + DOX 1 µg/mL and the plate was incubated to allow the formation of tumor spheroids. In this assay, the AFAT-Se-PTX-PCL-77KS-NP nanoformulation was essentially used, which showed promising results in the 2D model, as well as the association of free active compounds (AFAT-Se-PTX-Free) for comparative purposes. The concentrations of active compounds, regardless of whether they were co-encapsulated or in associated free form, were 15, 30, and 60 μg/mL for AFAT-Se and 3, 6, and 12 μg/mL for PTX. That is, the ratio was 15:3, 30:6, and 60:12 μg/mL (AFAT-Se:PTX). After the treatment application, the spheroids were incubated for 12 days, and photographed for monitoring and subsequent measurement. Size was determined using ImageJ/Fiji software (ImageJ 1.53k, National Institute of Health, Bethesda, MD, USA). Results are expressed as spheroid area percentage (%) determined in comparison to day 0, which was set as 100%. The following equation was used for the calculations.
$$\%Spheroids\;area=\left[\frac{Sphreroids\;area\;t\times 100}{Sphreroids\;area\;t0}\right]$$

### 2.13. Statistical Analyses

Recorded results are expressed as mean ± standard error (SE) or mean ± standard deviation (SD). Statistical analyses were conducted using one-way analysis of variance (ANOVA) to discern variations between data sets, followed by the Tukey or Duncan post hoc test for subsequent multiple comparisons. SPSS^®^ software (SPSS Inc., Chicago, IL, USA, version 22) was used for these statistical analyses. The robustness of the RP-LC method was verified using Minitab 17 (MINITAB^®^ Statistical Software, Release 17, Minitab Inc., State College, PA, USA). Each experimental iteration was replicated three times, and statistical significance was determined at a threshold of *p* < 0.05.

## 3. Results

### 3.1. Characterization of the NPs and Entrapment Efficiency

The physicochemical characterization of the nanoformulations is shown in Table 1. It contains data on the mean hydrodynamic diameter, PDI, ZP, and pH of the NPs. It is also worth mentioning that the entrapment efficiency (EE%) of both active compounds in all proposed formulations was around 99% (AFAT-Se 99.88% ± 0.5 and PTX 99.80% ± 0.6). The pH approaches the physiological environment, ~6.7, compatible with parenteral administration [26]. No significant difference was observed between the formulations (*p* > 0.05). Moreover, the SEM analysis showed that the co-loaded NPs have a roughly spherical shape (Figure 1), with particle size corroborating the values determined by DLS.

### 3.2. Analytical Method

A HPLC method was efficiently developed and validated for the simultaneous analysis of AFAT-Se and PTX in the NPs. The specificity assessment was carried out by analyzing the chromatograms (Figure 2) of the nanoformulation without the active compounds in comparison with NPs containing the compounds. Therefore, it can be highlighted that no component of the formulation interferes with the analysis of AFAT-Se and PTX at the detection wavelength of 227 nm [27].

The linear regression analysis and the correlation coefficient for the organoselenium compound AFAT-Se evidenced the method linearity in the range 1–50 μg/mL (y = 54,753x − 5870.5, r = 0.9999). In the same way, ANOVA analysis showed for the organoselenium that the method has a significant linear regression (Fcalculated = 14,270.83 > Fcritical = 4.75, *p* < 0.05), without linearity deviation (Fcalculated = 0.10 < Fcritical = 3.26, *p* > 0.05) [27].

For PTX, the method linearity was evidenced in the range 0.5–20 μg/mL (y = 33,733x + 6473.1, r = 0.9999). Furthermore, ANOVA analysis showed that the method has a significant linear regression (Fcalculated = 14,843.40 > Fcritical = 4.75, *p* < 0.05), without linearity deviation (Fcalculated = 0.44 < Fcritical = 3.26, *p* > 0.05) [27].

Evaluating the repeatability for both active compounds intraday and between analysts, the relative standard derivation (RSD) values were less than 2%, characterizing the precision of the proposed method, as well as the accuracy performed by the recovery test, which presented values between 98 and 102% (AFAT-Se 100.17% and PTX 100.11%). Moreover, the Pareto chart reveals that different adjustments made within the experimental conditions, such as variation of injection volume, and modification of the % of methanol and acetonitrile in the mobile phase, have no discernible impact on the assay performance (*p* > 0.05), evidencing the robustness of the analytical method [27].

### 3.3. Scavenging Activity

The scavenging activity was determined by evaluating AFAT-Se-PTX-Free, AFAT-Se-PTX-PCL-77KS-NP, and PCL-77KS-NP (Figure 3). It is noteworthy that in both assays the nanoencapsulation of the active compounds was able to maintain the antioxidant activity observed in the association of free compounds, especially at higher concentrations.

### 3.4. In Vitro Safety Profile

AFAT-Se-PTX-Free, AFAT-Se-PTX-PCL-77KS-NP, and PCL-77KS-NP did not demonstrate cytotoxic effects in the assay using the PBMC cell line. In the same sense, when evaluated in the 3T3 cell line, the lowest viability found was around ~73% at the highest concentrations of the test, evidencing the biocompatibility of the NPs at the lower tested concentrations, and only mild toxicity at the higher concentrations (Figure 4).

### 3.5. Hemocompatibility Studies

For nanoparticle systems designed for systemic administration, blood compatibility is an important parameter to be evaluated (Figure 5). Therefore, erythrocyte damage was measured through hemoglobin quantification, in which AFAT-Se-PTX-PCL-77KS-NP showed hemolytic percentages close to zero even at the highest concentrations of the assay. On the other hand, PCL-77KS-NP showed hemolysis rates (~10%) that indicate a moderate hemolytic formulation, especially at the highest concentrations of the assay. Moreover, it is worth noting that the association of free active compounds reached 58.57% hemolytic potential, evidencing the greater biocompatibility of the nanoformulations to overcome unspecific toxicity of free drugs [19].

### 3.6. pH-Dependent Membrane-Lytic Activity of Nanoparticles

The evaluation of the pH-dependent behavior conferred by the surfactant 77KS on the nanoformulations was carried out through a hemolysis assay [18]. Formulation without active compounds and the co-encapsulated NP showed similar behavior, since the presence of 77KS in both formulations was able to significantly increase the lytic capacity of the membrane when at pH 5.4, compared to pH 6.6 and 7.4. In contrast, the formulations without 77KS did not display a pH-responsive membrane-lytic behavior (Figure 6).

### 3.7. Synergic In Vitro Antitumor Activity Using 2D Cell Model

A resistant/MDR cell line, NCI/ADR-RES, was used to evaluate the antitumor activity and potential to overcome MDR of the two bioactive compounds, AFAT-Se and PTX, both in their free and nanoencapsulated forms [7,28,29]. Thus, it can be seen in Figure 7 that the nanoencapsulation of the free active ingredients, AFAT-Se-PCL-77KS-NP and PTX-PCL-77KS-NP (Figure 7A and 7B, respectively), was already able to significantly reduce cell viability at most of the tested concentrations. In the same way, the association of free compounds, AFAT-Se-PTX-Free, was more efficient in presenting cytotoxicity when compared to the non-associated compounds. Finally, the effect of the co-loaded NPs, AFAT-Se-PTX-PCL-77KS-NP, was noteworthy, and these further improved the antitumor potential of the bioactive compounds, AFAT-Se and PTX (Figure 7C).

The synergism of the association of free compounds (Figure 7D) occurred at all concentrations tested at different levels (CI < 1.0). The concentration of AFAT-Se-PTX-Free 1 (A) + 0.2 (P) presented an IC value of 0.07531 (very strong synergism, 0–0.1), whereas at concentrations of 3 (A) + 0.6 (P) and 5 (A) + 1 (P), IC values of 0.13471 and 0.28683 were found, respectively (strong synergism, 0.1–0.3). Finally, at the concentrations of 15 (A) + 3 (P) and 30 (A) + 6 (P), an IC value of 0.49615 and 0.36652 and was found (synergism, 0.3–0.7) [20,21,22].

The co-encapsulated nanoformulation showed an additive effect in the lowest concentration of the test, and notable synergistic effects in the remaining concentrations (Figure 7D). Therefore, AFAT-Se-PTX-PCL-77KS-NP at concentrations of 3 (A) + 0.6 (P) and 5 (A) + 1 (P) showed IC values of 0.49903 and 0.67498, respectively (synergism 0.3–0.7). Likewise, at the concentration of 15 (A) + 3 (P), an IC value of 0.19719 (strong synergism, 0.1–0.3) was observed, whereas at the concentration of 30 (A) + 6 (P), a value of 0.01973 (very strong synergism, 0–0.1) was evidenced. Antagonistic effects were not observed under the conditions used in this test [20,21,22].

### 3.8. In Vitro Antitumor Activity Using 3D Cell Models

Cultures based on three-dimensional models have enhanced predictability regarding efficacy and toxicity, given their ability to better preserve relevant aspects of cancer physiology. This approach is characterized by vital features such as cell–cell and cell–extracellular environment interactions, a tissue-like organization, and variable access to oxygen, nutrients, metabolites, and molecules. Therefore, the tumor spheroids are a 3D cell model that closely mimics the tumor cellular organization [30,31].

Highlighted in Figure 8 is the prominent ability of the association of the bioactive compounds to reduce cell viability at all tested concentrations over the days of analysis, with no significant difference between association of free compounds and the co-loaded formulation (*p* > 0.05). However, while at the end of 12 days of analysis the control reached 115.0% ± 2.9 of area (in comparison to day 0, which was set as 100%), AFAT-Se-PTX-PCL-77KS-NP at the highest concentration tested significantly decreased spheroid growth to 57.6% ± 0.7, with also a significant difference from the association of the free bioactive compounds, AFAT-Se-PTX-Free, which reduced the spheroid growth to 76.3% ± 4.8. Therefore, the nanoencapsulated-combined treatment was able to inhibit 57.4% of the growth rate with regard to the control spheroids, while the association of the free compounds inhibited only 38.7%. A notable difference in spheroid growth is visually perceived when analyzing the treatments in comparison to the untreated control in terms of the spheroid size following the days of analysis (Figure 9).

## 4. Discussion

Although progress has been made in oncological research, new treatments and improvements of conventional therapies are still required to reduce side effects and obtain tumor-specific treatments [32]. The most common treatment for cancer continues to be chemotherapy, despite having several associated complications and resistance mechanisms such as genetic factors, increased drug efflux, increased metabolism of xenobiotics, growth factors, and increased DNA repair capacity [7]. Some strategies may be important to circumvent the resistance mechanisms in cancer therapies and promote more efficient therapies. The AFAT-Se compound and other zidovudine derivatives were previously evaluated in antiproliferative assays on the T24 human bladder carcinoma cell line, demonstrating 76% growth inhibition after 48 h of incubation [12]. Therefore, considering the promising activity of AFAT-Se against sensitive tumor cell lines, here we proposed a combined therapy with the antitumor drug PTX, aiming to achieve an effective synergistic activity on resistant/MDR cells (NCI/ADR-RES cell line). Another approach used in this study was the co-encapsulation of the bioactive compounds in a pH-responsive nano-based system, aiming to achieve greater antiproliferative activity, likely due to an effective accumulation in tumor sites and intracellular compartments.

Nanotechnology and nanomedicine have emerged as promising areas in the quest for more effective treatments. Nanoscale-based delivery systems enable the possibility of enhancing pharmacokinetics, specificity, and circumvention of significant MDR mechanisms. Several nanopharmaceuticals are already on the market and have been approved by the FDA, while studies continue in the perspective of validating nanoparticles with this type of targeting [33]. In this context, monoclonal antibodies stand out among active transport strategies, offering advantages by binding to specific targets/antigens on tumor cells, enabling targeted delivery, and reducing toxicity [34]. This approach seems to be a potential strategy to functionalize the NPs proposed in our study and, thus, the design of antibody-targeted NPs stays as a future perspective.

The NPs under study were successfully obtained using the nanoprecipitation method, especially the main formulation, AFAT-Se-PTX-PCL-77KS-NP. Particle size and PDI were found to be satisfactory for the type of formulation proposed, and the SEM analysis enabled an effective visualization of the nanoformulation morphology. The hydrodynamic size range of NPs between 100 and 200 nm has been recognized as ideal for drug delivery systems due to specific advantages. The phenomenon of the enhanced permeability and retention (EPR) effect in tumors is remarkable, allowing NPs to exploit leaky vasculature and poor lymphatic drainage, and selectively accumulate in tumor tissues, thereby increasing the efficacy of drug delivery and minimizing adverse effects on healthy tissues. Additionally, this size range can prevent excessive filtration by the spleen, prolonging circulation time in the bloodstream and increasing the likelihood of reaching the target location. Likewise, absorption by the liver is impaired, ensuring that a greater proportion of the drug is available for targeted therapies [3,35]. The negative zeta potential is likely conferred by the charge of the polymer used, PCL; however, the low modulus value is noticeable and this is probably due to the presence of non-ionic surfactants (Pluronic^®^ F-127 and Span^®^ 80) possibly adsorbed on the surface. The presence of poloxamer is important in trying to circumvent the reticuloendothelial system (RES) and mononuclear phagocytic system (MPS), as well as involvement by macrophages. NPs with poloxamer already showed reduced systemic clearance and, consequently, prolonged circulation time due to the reduction in electrostatic and/or hydrophobic interaction with opsonins (endogenous plasma proteins). Additionally, poloxamer is capable of inhibiting P-gp, thus being able to reduce the phenomenon of efflux of active compounds and reducing MDR effects [36]. The encapsulation efficiency was successfully determined by the developed RP-LC method and was extremely satisfactory for both active ingredients. Notably, a high EE% is likely to prevent premature release of the loaded bioactive, inhibit blood deterioration, reduce the side effects due to unspecific action in healthy tissues and, ultimately, refine both pharmacodynamics and pharmacokinetics [37].

Selenium, considered an essential trace element, has been extensively explored when integrated into organic molecules. The promising antioxidant activity of such compounds, that is, the capture of free radicals and reactive oxygen species (ROS), justifies the exploration of this mechanism in this study [29,38]. The association of the free active compounds (AFAT-Se-PTX-Free) and the AFAT-Se-PTX-PCL-77KS-NP formulation were evaluated by the complementary ABTS and DPPH methods, with both assays capable of identifying potential antioxidants based on free radical capture. The increased sensitivity observed in the ABTS assay is possibly due to accelerated reaction kinetics coupled with an elevated responsiveness to antioxidants [39]. The results of good antioxidant potential found in this study corroborate those found by Leal and co-workers for the free AFAT-Se compound [12]. It is also worth noting that the remarkable antioxidant activity of the formulation may be attributed to the ferulic acid portion present in the molecule under study [40].

Oxidative stress in the tumor microenvironment is evident; thus, the generation of ROS is characteristic of cancer cells. Although high concentrations of ROS are cytotoxic and can exert antitumorigenic effects by causing oxidative damage and initiating ROS-dependent cell death signaling, these ROS also fulfill crucial roles in tumorigenesis and cancer progression. The formation of tumors or metastases can be influenced by the oxidative stress and allow damage to macromolecules, such as nucleic acids, proteins, lipids, and glucose, even resulting in genetic mutations and activation of pro-oncogenic signaling [41,42,43]. Therefore, bioactive compounds with an antioxidant mechanism could represent an important strategy for improving anticancer therapy.

The in vitro determination of the initial safety profile of the association of free compounds (AFAT-Se-PTX-Free), as well as of the co-loaded (AFAT-Se-PTX-PCL-77KS-NP) and unloaded (PCL-77KS-NP) nanoformulations, was carried out using assays involving PBMCs and the 3T3 cell line to ensure a more comprehensive analysis. It is important to note that these non-tumor cell models were used to initially identify and understand the nonspecific cytotoxic effects and, thus, the safety profile of the proposed nanoformulation. Later, a resistant tumor cell line (NCI/ADR-RES) was chosen to detect the antitumor potential of the NPs, especially regarding their capacity to overcome the MDR effect on cancer. A comparative study using non-tumor and tumor cells presents a notable relevance to identify the selective cytotoxicity of an innovative therapy towards cancer cells [44]. In the PBMC assay, cell viability remained >80% for all tested samples across the entire concentration range. In the assay using the 3T3 cell line, the lowest viability observed was approximately ~73%, indicating low cytotoxicity. In contrast, at the same concentration in the resistant tumor cell line, cellular viability was around 31%. Moreover, in line with these data, the hemolysis study corroborates the biocompatibility of the NPs. AFAT-Se-PTX-PCL-77KS-NP was shown to be non-hemolytic even at the high concentrations used in this trial; in contrast, the association of the free bioactive compounds resulted in considerable hemolysis rates in all tested concentrations.

The impact of different pH conditions on the ability of NPs to disrupt the membrane lipid bilayer was evaluated with human erythrocytes as a representative model for endosomal membranes. The pronounced membranolytic efficacy observed in formulations with 77KS became particularly apparent as the ambient pH decreased, especially at pH 5.4. On the other hand, NPs without 77KS exhibited an absence of this pH-responsive behavior, highlighting the fundamental role of 77KS on pH stimuli. A primary hypothesis indicates the protonation of the carboxylic group within the surfactant, allowing greater proximity to neutrality and the adoption of a lipophilic configuration. This, in turn, increases its ability to interact with the membrane, induce changes in permeability, and trigger lysis [5,14,18]. In this way, the presence of this pH-responsive surfactant in the NP matrix would facilitate endosomal disruption, thus promoting intracellular accumulation, which could help the active ingredients to easily reach their target inside the cell [14].

Cancer therapy based on nanometric formulation provides an excellent platform for combined therapy and overcoming unwanted effects caused by existing treatments [45]. Here, the cytotoxicity in the 2D assay toward the resistant/MDR cells of AFAT-Se-PCL-77KS-NP and PTX-PCL-77KS-NP evidenced the potential antitumor activity of the formulations separately containing the active ingredients. In contrast, the free compounds were not able to reduce the cell viability, indicating their negligible ability to overcome MDR. The innovative organoselenium AFAT-Se would possibly have a mechanism of action based on those found for nucleoside analogues, that is, the prevention of intracellular enzymes, such as polymerases or ribonucleotide reductase, or the prevention of the elongation of the DNA chain, through integration into it [4,5,46]. In the same way, the antitumor paclitaxel is a microtubule stabilizer, which induces mitosis arrest by interfering with spindle formation, thus resulting in cell death [6].

Cancer monotherapy, although still very common, can be overcome by combined therapy. Its superiority can be justified due to its action in multiple mechanisms, facilitating the regression of resistance effects, as cancer cells have difficulty preventing the toxic effects generated by two bioagents. Furthermore, slowing the growth of tumors, mitigating their potential to spread, and overcoming drug resistance, are critical goals in cancer treatment. Achieving success in these areas contributes to a greater likelihood of patient survival [8]. Therefore, here we studied the association of free and co-encapsulated active compounds. It was observed that the association of free compounds was quite effective, demonstrating synergistic effects at all concentrations, while, via the co-loaded NPs, the cytotoxic effects were more prominent, resulting in lower cell viability rates, and additive and synergistic effects were observed. Therefore, the combined treatment with AFAT-Se and PTX, especially when co-encapsulated into pH-sensitive PCL NPs, is capable of sensitizing the resistant/MDR tumor cells, achieving 68.5% cytotoxicity, as detected by the MTT assay in the highest concentration of the assay for AFAT-Se-PTX-PCL-77KS-NP.

The overexpression of the pg-P efflux pump is one of the factors responsible for resistance in MDR cells. Therefore, the observed synergism may also be corroborated by a preliminary in silico evaluation using quick, reliable, and free-access platforms such as pkCSM and admetSAR, in which the AFAT-Se compound was shown to be an inhibitor of this pump. In addition, the nanoformulation composition contains poloxamer 407, an adjuvant that also can assist in this inhibition and minimize MDR effects [5,36,47,48,49,50].

Cell cultures based on two-dimensional platforms are easy to handle and have the possibility of a large number of analyses; therefore, they are important in drug screening, but limitations exist [51]. Hence, the three-dimensional approach aims to improve cell–cell contact, providing a more reliable environment for cancer, with related abilities, reduced oxygen levels, restricted molecular movement, acidic interstitial environment, and alterations in metabolism [52]. Here, by applying the spheroids/3D cell model, it was highlighted that the association of free and co-encapsulated compounds was able to reduce the size of the spheroids over the course of the days of analysis when compared to controls, which have a considerable growth. The observed results evidenced that some concentrations were able to stop tumor growth, while others were responsible for the significant narrowing of the initial spheroid. However, the significant difference for the AFAT-Se-PTX-PCL-77KS-NP nanoformulation compared to AFAT-Se-PTX-Free found in the 2D tests was not as evident in the 3D test, demonstrating the greater resistance of this type of platform [29]. On the other hand, it is worth highlighting that in the highest concentration of the test, the co-loaded nanoformulation was still superior on the last day of the assay. Therefore, the 2D and 3D results evidenced that the innovative organoselenium AFAT-Se is a promising antitumor agent, especially when associated with PTX. In addition, its encapsulation in a nanoformulation, the co-loading with PTX, together with the pH-dependent behavior of the NP, may result in a pharmacological approach to improve therapies and overcome resistance/MDR effects in the cancer.

The overall results of this study are promising, but it is indeed important to recognize the limitations for future advancements, given the significance of the innovative combination proposed here. The employed cellular models are well-recognized and commonly used; however, they may not fully capture the in vivo context and complexity, thus limiting the generalization and immediate correlation of the results, both in terms of activity and toxicity in a clinical context. Additional studies, including animal models, might be the next steps to take. Long-term effects of the nanoformulation could also be of interest, including assessments involving its role on platelet function, as well as studies of its impact on the immune system and cytokine expression in the tumor microenvironment.

## 5. Conclusions

This study underscores the continued need for advances in oncology research to develop new therapies and improve conventional treatment approaches, with the goal of mitigating side effects and providing targeted treatments. The focus on combining the organoselenium AFAT-Se with PTX in a pH-responsive nanoparticulate system has shown promising results, demonstrating synergistic activity against resistant/MDR cancer cells. The successful formulation of AFAT-Se-PTX-PCL-77KS-NP, characterized by its size and PDI, has potential for drug delivery by exploiting the EPR effect in tumors. The comprehensive analysis, including the safety assessments, and 2D and 3D in vitro tumor cell culture experiments, highlights the biocompatibility of the NPs, and the effectiveness of the combination of these bioactive compounds to promote cell viability decrease in resistant/MDR cells, especially when co-encapsulated. Moreover, the antioxidant activity of the proposed nanodevice can also help prevent tumorigenesis. The overall results suggest that this innovative approach, incorporating the organoselenium AFAT-Se and PTX in a pH-responsive nanoparticulate system, represents a valuable strategy for overcoming multidrug resistance and improving cancer therapies.

## Figures and Tables

**Figure 1 pharmaceutics-16-00590-f001:**
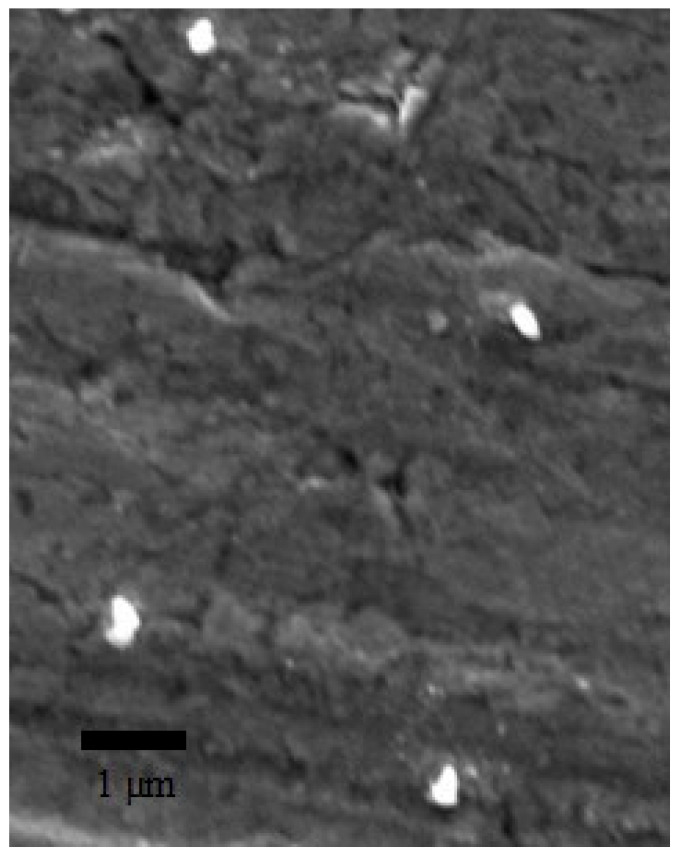
AFAT-Se-PTX-PCL-77KS-NP morphology by SEM.

**Figure 2 pharmaceutics-16-00590-f002:**
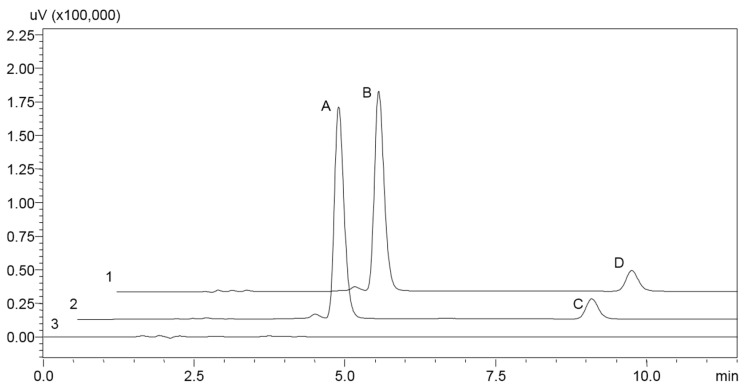
Chromatograms obtained after method validation for the active compounds and nanoformulations under analysis. 1—AFAT-Se-PTX-Free, 2—AFAT-Se-PTX-PCL-77 KS-NP, and 3—PCL-77KS-NP. Peaks A and B correspond to AFAT-Se; peaks C and D correspond to PTX.

**Figure 3 pharmaceutics-16-00590-f003:**
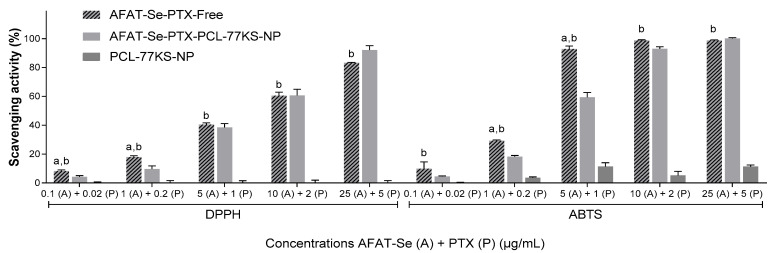
Scavenging activity of AFAT-Se-PTX-Free, AFAT-Se-PTX-PCL-77KS-NP, and PCL-77KS-NP using DPPH and ABTS assays. Results are expressed as mean ± SE of three independent experiments. Statistical analyses were performed using ANOVA followed by the Tukey post hoc test. ^a^ Significantly different from AFAT-Se-PTX-PCL-77KS-NP (*p* < 0.05), ^b^ from PCL-77KS-NP (*p* < 0.05).

**Figure 4 pharmaceutics-16-00590-f004:**
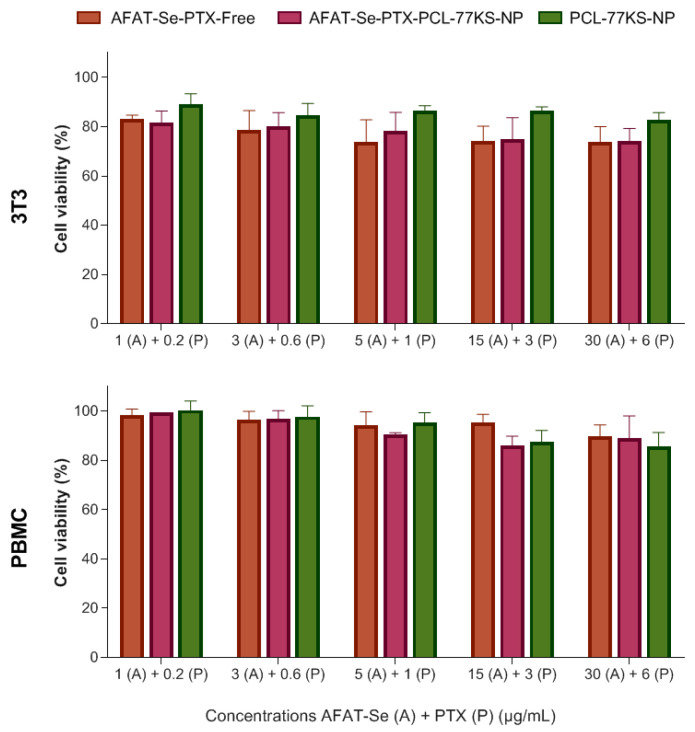
Safety profile using non-tumor cell line 3T3 and human mononuclear cells of peripheral blood (PBMC). Results are expressed as mean ± SE of three independent experiments. Statistical analyses were performed using ANOVA followed by the Tukey post hoc test. No significant differences were found between concentrations used in the assays.

**Figure 5 pharmaceutics-16-00590-f005:**
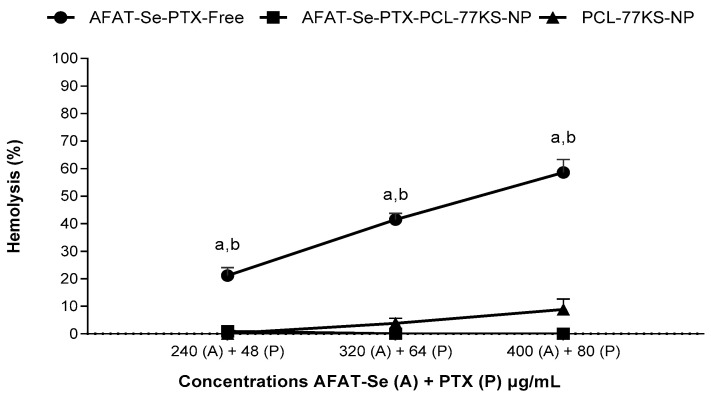
Hemocompatibility study of AFAT-Se-PTX-Free, AFAT-Se-PTX-PCL-77KS-NP, and PCL-77KS-NP after incubation with human erythrocytes. Results are expressed as mean ± SE of three independent experiments. Statistical analyses were performed using ANOVA followed by the Tukey post hoc test. ^a^ Significantly different from AFAT-Se-PTX-PCL-77KS-NP (*p* < 0.05), ^b^ from PCL-77KS-NP (*p* < 0.05).

**Figure 6 pharmaceutics-16-00590-f006:**
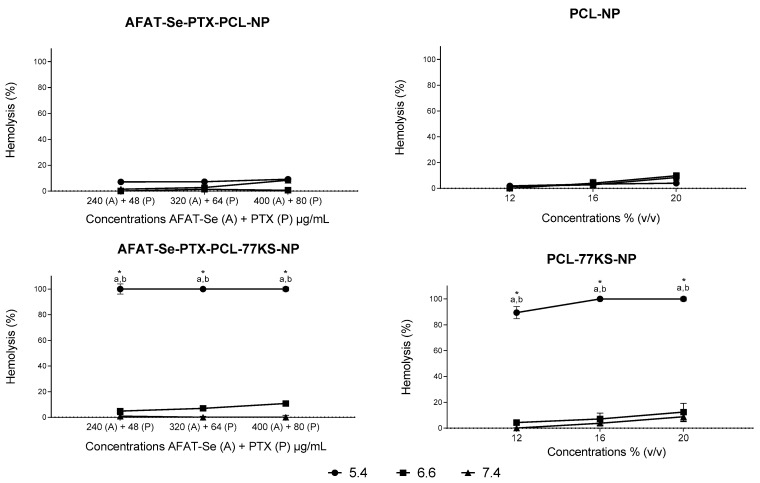
pH-dependent membrane-lytic activity of AFAT-Se-PTX-PCL-NP and PCL-NP (NPs without 77KS), and AFAT-Se-PTX-PCL-77KS-NP and PCL-77KS-NP (NPs with 77KS), after 5 h of incubation with human erythrocytes at pH 5.4, 6.6, and 7.4. Each value represents mean ± SE of three experiments. Statistical analyses were performed using ANOVA followed by the Tukey post hoc test. ^a^ Significantly different from pH 7.4 (*p* < 0.05) and ^b^ from pH 6.6 (*p* < 0.05). * Indicates significant difference between formulations with and without 77KS (*p* < 0.05).

**Figure 7 pharmaceutics-16-00590-f007:**
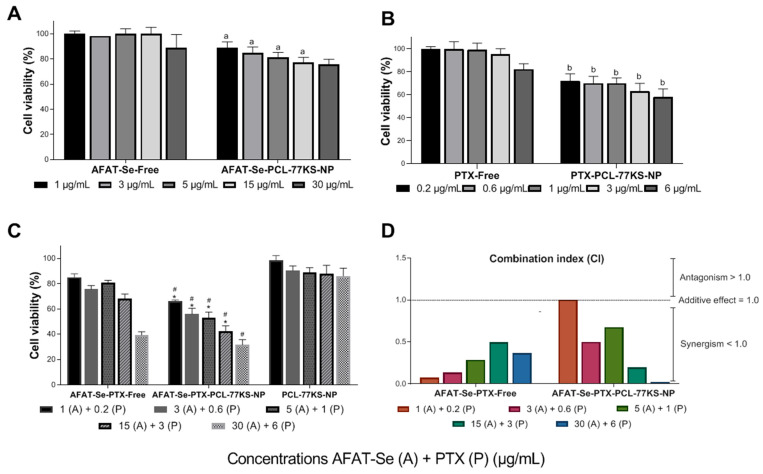
(**A**–**C**) In vitro cell viability in NCI/ADR-RES cell line using 2D cell model by MTT assay after 72 h of treatment. Data are expressed as mean of three independent experiments ± SE. Statistical analyses were performed using ANOVA followed by the Duncan post hoc test. ^a^ significant difference from AFAT-Se-Free (*p* < 0.05), ^b^ significant difference from PTX-Free, * significant difference from AFAT-Se-PTX-Free and ^#^ significant difference from PCL-77KS-NP. (**D**) Combination index values for the association of active compounds, AFAT-Se-PTX-Free and AFAT-Se-PTX-PCL-77KS-NP, in NCI/ADR-RES cell line.

**Figure 8 pharmaceutics-16-00590-f008:**
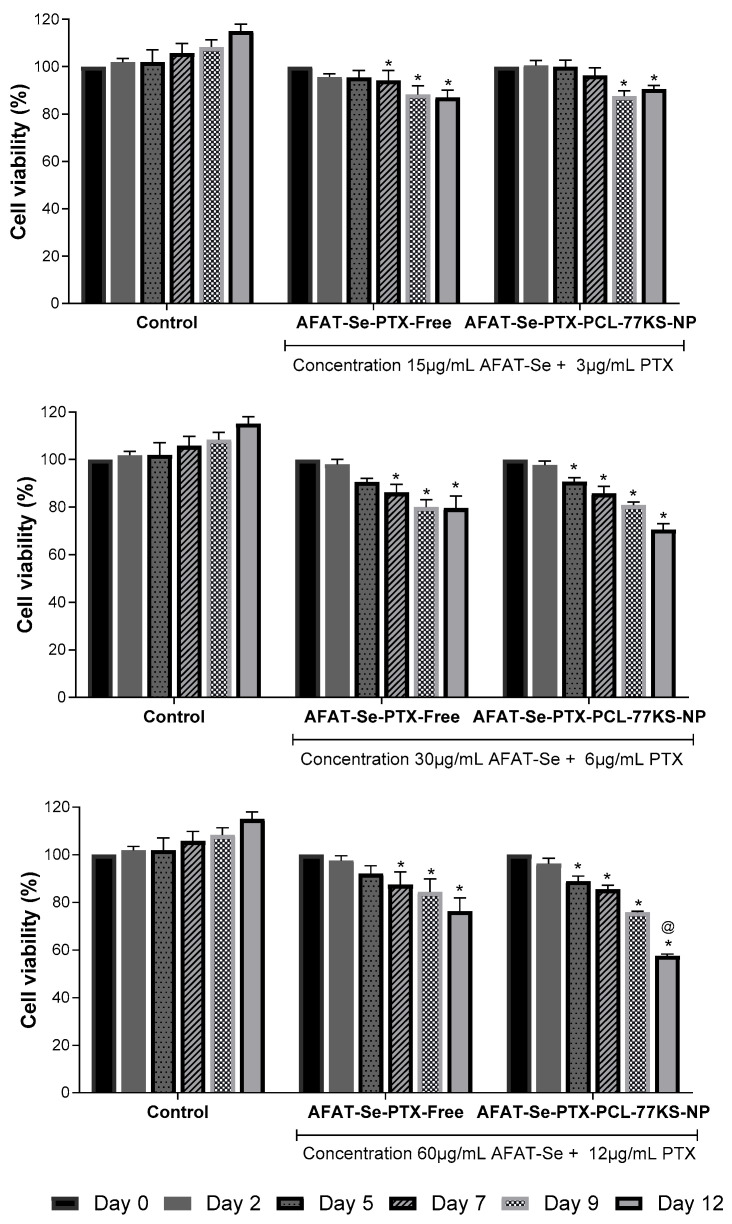
Cytotoxicity of AFAT-Se-PTX-Free and AFAT-Se-PTX-PCL-77KS-NP against NCI/ADR-RES spheroids. Spheroid area percentage (%) determined in comparison to day 0, which was set as 100%. Statistical analyses were performed using ANOVA followed by the Tukey post hoc test. * significant difference from control, ^@^ significant difference from AFAT-Se-PTX-Free.

**Figure 9 pharmaceutics-16-00590-f009:**
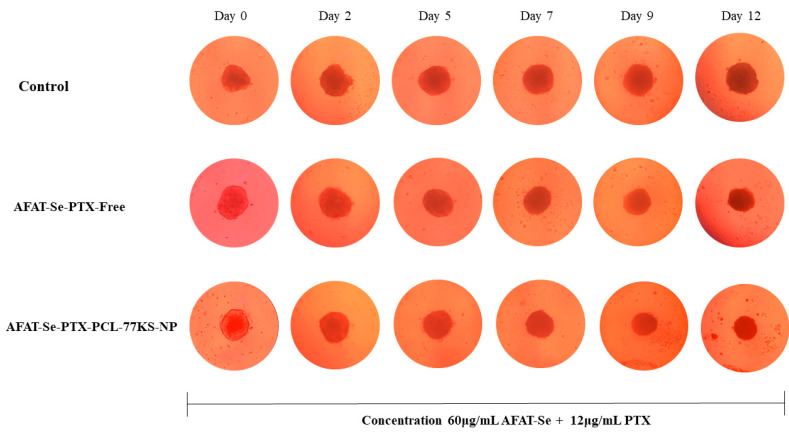
Representative images of NCI/ADR-RES spheroids treated with AFAT-Se-PTX-Free and AFAT-Se-PTX-PCL-77KS-NP at concentration of 60 μg/mL AFAT-Se + 12 μg/mL PTX. Images were obtained using an inverted microscope at day 0 (before treatment), and after exposure to the treatments for 2, 5, 7, 9, and 12 days.

**Table 1 pharmaceutics-16-00590-t001:** Physicochemical characterization of NPs.

	Particle Size (nm) ± SD	PDI ± SD	ZP (mV) ± SD	pH ± SD
AFAT-Se-PTX-PCL-77KS-NP	173.3 ± 0.9	0.082 ± 0.006	−3.6 ± 0.2	6.7 ± 0.3
AFAT-Se-PCL-77KS-NP	164.8 ± 2.0	0.110 ± 0.013	−3.66 ± 0.1	6.6 ± 0.4
PTX-PCL-77KS-NP	169.3 ± 2.9	0.095 ± 0.001	−3.37 ± 0.5	6.8 ± 0.2
PCL-77KS-NP	167.1 ± 4.1	0.079 ± 0.004	−3.88 ± 0.7	6.6 ± 0.1

SD, standard deviation, n = 3.

## Data Availability

Data are contained within the article.

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
