# Peer review of "Co-Delivery of an Innovative Organoselenium Compound and Paclitaxel by pH-Responsive PCL Nanoparticles to Synergistically Overcome Multidrug Resistance in Cancer"

_pharmaceutics, 2024, doi:10.3390/pharmaceutics16050590_

Round 1
Reviewer 1 Report
Comments and Suggestions for Authors
The authors of the paper entitled "Co-delivery of an innovative organoselenium compound and paclitaxel by pH-responsive PCL nanoparticles to synergistically overcome multidrug resistance in cancer" provide an elegant way for delivering AFAT-Se & paclitaxel (PTX) for overcoming multidrug resistance.
The presented results demonstrates impressive entrapment efficacy with little wastage of the drug. The NPs thus produced also seem to be quite very robust activity in vitro. However the authors need to address a few aspects before article can be published.
Some of the aspects that need the attention for the authors are as indicated below:
Introduction - please include rationale for combining paclitaxel with AFAT-Se. It is metioned in discussion, kindly include it to introduction as well
Small gramatical errors such as -
line 403 - change to "did not display" from "did not displayed"
Paclitaxel confers considerable thrombocytopenia, the effects of these nanoparticles evaluated only effects on RBC's. Kindly perform Platelets evaluation as well.
The authors tend to refer to data that has not been published couple of times. Suggest the following changes:
Line 492 - 494.
Line 604 - 608.
The authors should provide evidence for their claim. If however a seperate paper is being prepared for communication, kindly remove these section
There has been quite a number pH dependent NP formulations that are being published. THe authors would benefit from showing in vivo efficacy. It is strongly suggested the authors perform an in vivo experiment to demonstrate the efficacy of this formulation
Comments on the Quality of English LanguageThe authors of the paper entitled "Co-delivery of an innovative organoselenium compound and paclitaxel by pH-responsive PCL nanoparticles to synergistically overcome multidrug resistance in cancer" provide an elegant way for delivering AFAT-Se & paclitaxel (PTX) for overcoming multidrug resistance.
The presented results demonstrates impressive entrapment efficacy with little wastage of the drug. The NPs thus produced also seem to be quite very robust activity in vitro. However the authors need to address a few aspects before article can be published.
Some of the aspects that need the attention for the authors are as indicated below:
Introduction - please include rationale for combining paclitaxel with AFAT-Se. It is metioned in discussion, kindly include it to introduction as well
Small gramatical errors such as -
line 403 - change to "did not display" from "did not displayed"
Paclitaxel confers considerable thrombocytopenia, the effects of these nanoparticles evaluated only effects on RBC's. Kindly perform Platelets evaluation as well.
The authors tend to refer to data that has not been published couple of times. Suggest the following changes:
Line 492 - 494.
Line 604 - 608.
The authors should provide evidence for their claim. If however a seperate paper is being prepared for communication, kindly remove these section
There has been quite a number pH dependent NP formulations that are being published. THe authors would benefit from showing in vivo efficacy. It is strongly suggested the authors perform an in vivo experiment to demonstrate the efficacy of this formulation
Author Response
The authors of the paper entitled "Co-delivery of an innovative organoselenium compound and paclitaxel by pH-responsive PCL nanoparticles to synergistically overcome multidrug resistance in cancer" provide an elegant way for delivering AFAT-Se & paclitaxel (PTX) for overcoming multidrug resistance.
The presented results demonstrates impressive entrapment efficacy with little wastage of the drug. The NPs thus produced also seem to be quite very robust activity in vitro. However the authors need to address a few aspects before article can be published.
Some of the aspects that need the attention for the authors are as indicated below:
- Introduction - please include rationale for combining paclitaxel with AFAT-Se. It is metioned in discussion, kindly include it to introduction as well
Authors: We thank the reviewer for this important suggestion. As recommended, the introduction section was adjusted, including the rationale for combining PTX and AFAT-Se, as follows:
Line 83: “Facing the limitations of individual treatments and the potentialities of the nano-technology, here we proposed a combined therapy of PTX and a novel organoselenium nucleoside analogue, 5'-Seleno-(phenyl)-3'-(ferulic-amido)-thymidine (AFAT-Se), which has shown promising antitumor effects in a previous in vitro study [12]. The rationale of the combination of these bioactive compounds relies on the potential to achieve an effective treatment that could synergically overcome MDR in cancer cells. Using different drugs that act by different mechanisms of action is likely to achieve greater antineoplastic activity, sensitization of resistant tumor cells, and decrease of side effects due to the reduction of therapeutic doses of each drug. Combined with these advantages, the co-encapsulation of the bioactive compounds in a nano-based system is expected to result in further improvements in the therapeutic outcomes.”
- Small gramatical errors such as - line 403 - change to "did not display" from "did not displayed"
Authors: We thank the reviewer for the valuable comment. The text was adjusted as recommended:
Line 422: “In contrast, the formulations without 77KS, did not display a pH-responsive membrane lytic behavior (Figure 5).”
- Paclitaxel confers considerable thrombocytopenia, the effects of these nanoparticles evaluated only effects on RBC's. Kindly perform Platelets evaluation as well.
Authors: We appreciate the consideration and recognize the significance of the evaluations of platelets function. However, our laboratory currently lacks the necessary materials to conduct such analyses. It worth noting that the AFAT-Se-PTX-PCL-77KS-NPs were able to significantly increase the antitumor activity, in comparison to the single treatments, which suggests that this synergistic treatment approach is possible to reduce the therapeutic doses of the individual drugs. In this sense, the dose reduction would be likely to also reduce all the side effects of paclitaxel. Additionally, we also do not concern with thrombocytopenia effects induced by the NPs, since, usually those effects are more common in oppositely charged particles [1].
[1] JESWANI, Gunjan et al. Development and optimization of paclitaxel loaded Eudragit/PLGA nanoparticles by simplex lattice mixture design: Exploration of improved hemocompatibility and in vivo kinetics. Biomedicine & Pharmacotherapy, v. 144, p. 112286, 2021.
In order to consider the lack of this evaluation as a limitation of our study, we included the following paragraph in the discussion section of the revised version of the manuscript:
Line 665: “The overall results of this study are promising, but it is indeed important to recognize the limitations for future advancements, given the significance of the innovative combination proposed here. The employed cellular models are well-recognized and commonly used; however, they may not fully capture the in vivo context and complexity, thus limiting the generalization and immediate correlation of the results, both in terms of activity and toxicity in a clinical context. Additional studies, including animal models, might be the next steps to take. Long-term effects of the nanoformulation could also be of interest, including assessments involving its role on platelets function, as well as studies on its impact on the immune system and cytokine expression in the tumor microenvironment.”
- The authors tend to refer to data that has not been published couple of times. Suggest the following changes: Line 492 - 494. Line 604 - 608. The authors should provide evidence for their claim. If however a seperate paper is being prepared for communication, kindly remove these section
Authors: We appreciate your attention to this matter. In fact, a separate paper is being prepared covering in silico and in vitro effects of AFAT-Se. In this context, and following your suggestion, we removed the first data (Line 492-494), and adjusted the second sentence, summarizing the approaches used to find the mentioned results. We think that it is interesting to maintain these data in the present manuscript to better discuss our experimental results. Therefore, having your agreement, we suggested the following adjustment in the text:
Line 640: “…corroborated by a preliminary in silico evaluation using quick, reliable, and free access platforms such as pkCSM and admetSAR, in which the AFAT-Se compound was shown to be an inhibitor of this pump.”
- There has been quite a number pH dependent NP formulations that are being published. THe authors would benefit from showing in vivo efficacy. It is strongly suggested the authors perform an in vivo experiment to demonstrate the efficacy of this formulation
Authors: We thank the reviewer for the valuable comment and totally agree that in vivo studies would strengthen the overall research. We understand the relevance of in vivo data to corroborate the in vitro results; however, it is not simple to perform both in vitro (2D and 3D approaches) and in vivo tests in the same study, especially facing some limitations in the laboratory structure and lack of enough funding. Therefore, we can highlight that the goal of this manuscript is to focus on in vitro experiments, using a resistant tumor cell line arranged in 2D and 3D experimental approaches, aiming to study the antitumor activity and the synergic effect of the proposed nanoparticles. Nevertheless, due to the promising results, as a second part of this study, we aim to evaluate the in vivo performance of the AFAT-Se-PTX-PCL-77KS-NP. Therefore, having your agreement and comprehension, we would like to let it as a subsequent study.

Reviewer 2 Report
Comments and Suggestions for Authors
The authors mathes and colleagues report the development o fan innovative PCL-based nanosystem for the dual delivery of organoselenium compound and paclitaxel for overcoming multidrug resistance in solid tumors.
The work is interesting since it focus on a new drug delivery system.
The manuscript would benefit from the followings:
1. A graphical abstract underling the mechanism of action of this innovative drug delivery system should be included.
2. The nanosystem should be morphologically characterized through SEM analysis
3. In order to corroborate the in vitro data the cytotoxic effect of the formulation should be
implemented with FACS or Tunel analysis.
4. A great variety of drug delivery systems has been proposed for solid lesions: in particular, the authors should discuss the possibility to couple a monoclonal antibody with chemotherapeutic agents within the same nanoparticle ( doi.org/10.1016/j.jconrel.2021.05.011).
Moreover, the increasing relevance of the role of nanomedicine in the landscape of solid malignancies should be highlighted. In this regard, recent advances in the field of rare tumors should be mentioned as reported in the following reference (doi.org/10.3389/fbioe.2022.953555).
5. Limitations of the study should be included.
Author Response
The authors mathes and colleagues report the development o fan innovative PCL-based nanosystem for the dual delivery of organoselenium compound and paclitaxel for overcoming multidrug resistance in solid tumors.
The work is interesting since it focus on a new drug delivery system.
The manuscript would benefit from the followings:
- A graphical abstract underling the mechanism of action of this innovative drug delivery system should be included.
Authors: AFAT-Se is a recently introduced compound, and its mechanism of action has not been fully elucidated. Similarly, the proposed synergistic association with a well-established antimitotic drug is also a novel concept. At this stage, we can only speculate that the mode of action of the organoselenium compound may bear similarity to analogous nucleoside molecules due to its molecular structure. This class is associated with inhibiting DNA polymerase or ribonucleotide reductase and the ability to insert themselves into DNA after phosphorylation. However, it is important to note that there is a scarcity of evidence to support this theory given the innovative nature of the proposed study. Therefore, having your agreement, we proposed a graphical abstract that focuses on the promising and evident results obtained for the formulation.
- The nanosystem should be morphologically characterized through SEM analysis
Authors: As recommended, the co-loaded nanoparticles (as the main NP formulation designed in this study) were morphologically characterized through SEM analysis. The methodology, results and the image of SEM was added in the revised version of the manuscript. This figure was included as Figure 1.
Line 172: “The morphology of the co-loaded NP (AFAT-Se-PTX-PCL-77KS-NP) was evaluated employing scanning electron microscopy (SEM) (JEOLJSM 6360, Akishima, Japan). In this process, 20 µL of the NP suspension was deposited onto a stub and incubated for 12 hours at ambient temperature. Subsequently, the stub was subjected to gold coating under diminished pressure, followed by examination of the samples utilizing a voltage of 10 kV.”
Line 336: “Moreover, the SEM analysis showed that the co-loaded NPs have a roughly spherical shape (Figure 1), with particle size corroborating the values determined by DLS.”
Line 342: “Figure 1. AFAT-Se-PTX-PCL-77KS-NP morphology by SEM.”
Line 533: “….of formulation proposed, and the SEM analysis enabled an effective visualization of the nanoformulation morphology.”
- In order to corroborate the in vitro data the cytotoxic effect of the formulation should be implemented with FACS or Tunel analysis.
Authors: We thank the reviewer for the valuable suggestion. The evaluation of FACS or Tunel analysis would certainly improve the interpretation of our antitumor activity results; however, at this moment we do not have the overall reagents and equipment necessary to perform these experiments in our lab. We understand the importance of additional experiments; nevertheless, we also want to highlight that this is the first study of the proposed NPs, and we intend to better elucidate the mechanisms underlying the cytotoxic effects as well as the effects of the NPs on in vivo models, in a subsequent study. Therefore, having your agreement and comprehension, we would like to perform these analyses in our further study.
- A great variety of drug delivery systems has been proposed for solid lesions: in particular, the authors should discuss the possibility to couple a monoclonal antibody with chemotherapeutic agents within the same nanoparticle (doi.org/10.1016/j.jconrel.2021.05.011).
Moreover, the increasing relevance of the role of nanomedicine in the landscape of solid malignancies should be highlighted. In this regard, recent advances in the field of rare tumors should be mentioned as reported in the following reference (doi.org/10.3389/fbioe.2022.953555).
Authors: We thank the reviewer for the suggestion of the references to improve the discussion of our study. However, the reference suggest regarding the monoclonal antibodies (doi.org/10.1016/j.jconrel.2021.05.011) do not correspond to this theme. The title of the article is “Heat/pH-boosted release of 5-fluorouracil and albumin-bound paclitaxel from Cu-doped layered double hydroxide nanomedicine for synergistical chemo-photo-therapy of breast cancer”, and the studies performed did not involve the couple of monoclonal antibodies within the nanosystem. In order to include the discussion in this context, we included other reference (10.2174/0929867327666200525161359), as reported below. The other suggested reference (doi.org/10.3389/fbioe.2022.953555) was cited in the revised version of the manuscript, as follows:
Line 521: “Nanotechnology and nanomedicine have emerged as promising areas in the quest for more effective treatments. Nanoscale-based delivery systems enable the possibility of enhancing pharmacokinetics, specificity, and circumvention of significant MDR mechanisms. Several nanopharmaceuticals are already on the market and have been approved by the FDA, while studies continue in the perspective of validating nanoparticles with this type of targeting [33]. In this context, monoclonal antibodies stand out among active transport strategies, offering advantages by binding to specific targets/antigens on tumor cells, enabling targeted delivery, and reducing toxicity [34]. This approach seems to be a potential strategy to functionalize the NPs proposed in our study and, thus, the design of antibody-targeted NPs stays as a future perspective.”
- Limitations of the study should be included.
Authors: As recommend, we included the limitations of our study at the end of the discussion section. The text was adjusted as follows:
Line 665: “The overall results of this study are promising, but it is indeed important to recognize the limitations for future advancements, given the significance of the innovative combination proposed here. The employed cellular models are well-recognized and commonly used; however, they may not fully capture the in vivo context and complexity, thus limiting the generalization and immediate correlation of the results, both in terms of activity and toxicity in a clinical context. Additional studies, including animal models, might be the next steps to take. Long-term effects of the nanoformulation could also be of interest, including assessments involving its role on platelets function, as well as studies on its impact on the immune system and cytokine expression in the tumor microenvironment.”

Reviewer 3 Report
Comments and Suggestions for Authors
This manuscript contains the outcomes of a study that involved a new anti-cancer drug development and testing for its anti-cancer effects. In this study, the authors produced, purified and characterized an effective synergistic combination of paclitaxel with a novel organoselenium nucleoside analogue, AFAT-Se. In addition, they observed that these bioactive compounds could effectively and satisfactorily be co-loaded into a nanoformulation containing a pH-dependent stabilizing surfactant (77KS) with increasing tumor sensitization, and the polymer poly (ε-caprolactone). Next, they tested their effects on cell viability/toxicity on cancer lines and human peripheral blood mononuclear cells (PBMC) in tissue culture as well in 2D conditions. They also tested the effects on 3D spheroids of resistant cancer cell lines. Furthermore, the authors determined the role of the pH on the membrane-lytic activity of the pH-sensitive NPs using the erythrocyte as a model for the endosomal membrane.
Also, the authors assessed the safety of their conjugated drugs via hemocompatibility assays and then determined the nonspecific cytotoxicity of the drugs using two non-tumor cell models. The anti-tumor activity, synergistic effects and the potential for overcoming the MDR effect were evaluated using NCI/ADR-RES resistant cell line by means of two-dimensional (2D) and three-dimensional (3D) in vitro platforms. They used combination of statistical approaches to analyze their results. Their methods for data presentation was also standard. Lastly, their conclusions were based on their observed results.
Overall, their results are indicative that their innovative approach of incorporating the organoselenium AFAT-Se and PTX in a pH-responsive nanoparticulate system, represents a valuable strategy for overcoming multidrug resistance and improving cancer therapies. The observed that their compounds are far less toxic to PBMC. From this study, they concluded that their approaches could represent a step forward in new anti-cancer drug development.
In general, it is an interesting study with very important scientific outcomes. The manuscript is nicely written with a few typo/grammatical errors, that can easily be corrected. The manuscript will be attractive to readers in the field and could be an inspirational for more of such studies to be designed for identifying new potential anti-cancer drugs. However, there are few questions to be addressed by the authors.
Questions:
1. In the initial viability/cytotoxic assay, you used 3T3 cell lines. However, in your 2D and 3D viability/toxicity assays you used different cell lines. Please, provide an explanation/clarification for readers.
2. You observed in your 3D viability assays that at highest concentration of AFAT-Se-PTX-PCL- 77KS-NP significantly decreased spheroid growth to 57.6% ± 0.7 relative to the spherical controls. Also, there was significant difference from the association of the free bioactive compounds, AFAT-Se-PTX-Free, tested, the lowest viability observed was 76. 3% in the resistant cell lines. These observations imply a significant loss in viability. However, based on these results, you concluded your conjugated compounds were relatively save, and may cause far less harm. Do you anticipate similar or identical results if you perform similar experiments using other cell lines such as the 3T3 or than resistant tumor cells?
3. Also, under the assumption that the 3D cell culture is relatively closer to in vivo than does the 2D does the percent toxicity caused by the highest concentration of your compounds could be a concern in an animal system. Do you see that as a potential problem? Please, offer your comment/response.
4. Your results support the possibility your compounds will be safe for cancer treatment. But did you determine its potential effects on immune system by examining their effects on T-cells viability/T cell activation?
5. Did you determine the effects of the conjugated compounds on cytokine production? This crucial because your compounds could induce expression and secretion of pro-inflammatory cytokine expression (IL-1beta, IL-6 and TNF), which could alter the tumor microenvironment and hence be detrimental to cancer patients who may be treated with these NPs.
6. If you did not determine the effects of your NPs on cytokine expression in cancer cell lines, it may be better to mention them in your discussions as potential limitations of your studies.
Minor concerns:
There are a few grammatical errors that can easily be corrected.
Examples: On line 525, the word determine should determined
Line 605, corroborate should corroborated
Line 618, growth should be growth
Comments on the Quality of English LanguageThere are a few grammatical errors that need to be corrected.
Author Response
This manuscript contains the outcomes of a study that involved a new anti-cancer drug development and testing for its anti-cancer effects. In this study, the authors produced, purified and characterized an effective synergistic combination of paclitaxel with a novel organoselenium nucleoside analogue, AFAT-Se. In addition, they observed that these bioactive compounds could effectively and satisfactorily be co-loaded into a nanoformulation containing a pH-dependent stabilizing surfactant (77KS) with increasing tumor sensitization, and the polymer poly (ε-caprolactone). Next, they tested their effects on cell viability/toxicity on cancer lines and human peripheral blood mononuclear cells (PBMC) in tissue culture as well in 2D conditions. They also tested the effects on 3D spheroids of resistant cancer cell lines. Furthermore, the authors determined the role of the pH on the membrane-lytic activity of the pH-sensitive NPs using the erythrocyte as a model for the endosomal membrane.
Also, the authors assessed the safety of their conjugated drugs via hemocompatibility assays and then determined the nonspecific cytotoxicity of the drugs using two non-tumor cell models. The anti-tumor activity, synergistic effects and the potential for overcoming the MDR effect were evaluated using NCI/ADR-RES resistant cell line by means of two-dimensional (2D) and three-dimensional (3D) in vitro platforms. They used combination of statistical approaches to analyze their results. Their methods for data presentation was also standard. Lastly, their conclusions were based on their observed results.
Overall, their results are indicative that their innovative approach of incorporating the organoselenium AFAT-Se and PTX in a pH-responsive nanoparticulate system, represents a valuable strategy for overcoming multidrug resistance and improving cancer therapies. The observed that their compounds are far less toxic to PBMC. From this study, they concluded that their approaches could represent a step forward in new anti-cancer drug development.
In general, it is an interesting study with very important scientific outcomes. The manuscript is nicely written with a few typo/grammatical errors, that can easily be corrected. The manuscript will be attractive to readers in the field and could be an inspirational for more of such studies to be designed for identifying new potential anti-cancer drugs. However, there are few questions to be addressed by the authors.
Questions:
- In the initial viability/cytotoxic assay, you used 3T3 cell lines. However, in your 2D and 3D viability/toxicity assays you used different cell lines. Please, provide an explanation/clarification for readers.
Authors: We appreciate your attention to this matter. The choice of cell lines was based on the context of each experiment and considerations related to the objectives of the assays conducted. The non-tumor 3T3 cell line was used in conjunction with PBMCs to initially identify and understand nonspecific cytotoxicity and safety considerations of the proposed nanoformulation, as these cell lines are well-established models for analyzing the effects of compounds on normal cells. On the other hand, in the assessment of antitumor activity, a resistant tumor cell line (NCI/ADR-Res) was used in different cell culture models to identify the formulation's action in causing cytotoxicity considering phenotypic characteristics of cancer cells and the context of resistance to multiple drugs. In order to provide an explanation/clarification for readers, the following sentence was included in the discussion section in the revised version of the manuscript:
Line 584: “It is important to note that these non-tumor cell models were used to initially identify and understand the nonspecific cytotoxic effects and, thus, the safety profile of the proposed nanoformulation. Later, a resistant tumor cell line (NCI/ADR-RES) was chosen to detect the antitumor potential of the co-loaded NP, especially regarding to its capacity to overcome the MDR effect on cancer. A comparative study using non-tumor and tumor cells presents a notable relevance to identify the selective cytotoxicity of an innovative therapy towards cancer cells.”
- You observed in your 3D viability assays that at highest concentration of AFAT-Se-PTX-PCL- 77KS-NP significantly decreased spheroid growth to 57.6% ± 0.7 relative to the spherical controls. Also, there was significant difference from the association of the free bioactive compounds, AFAT-Se-PTX-Free, tested, the lowest viability observed was 76. 3% in the resistant cell lines. These observations imply a significant loss in viability. However, based on these results, you concluded your conjugated compounds were relatively save, and may cause far less harm. Do you anticipate similar or identical results if you perform similar experiments using other cell lines such as the 3T3 or than resistant tumor cells?
Authors: Usually, 3D models are more resistant than 2D models, as cell organization in the spheroids leads to a lower surface area between the drug and cells. This higher resistance was evidenced in our results. Concerning the results achieved in the 3D assay, the high viability found for the association of the free compounds is related to its low antitumor potential in comparison to the co-loaded nanoformulation. Therefore, by using the 3D model, we are not assuming that the association of free compounds is safe, but we are concluding that it is less effective towards tumor cells than the co-loaded NP. Indeed, we suggest the safety of the NPs due to their significant antitumor activity in both NCI/ADR-RES 2D and 3D models (more resistant) and low cytotoxicity in non-tumor cells using the 2D model (more sensitive). Moreover, we can assume that NPs would also be non-cytotoxic in non-tumor cells (3T3) using a more resistant model (3D model). We did not perform a 3D assay using non-tumor cells, because the rationale of this 3D assay is to simulate a tumor tissue using in vitro conditions. The obtained spheroids are classified as tumorspheres.
- Also, under the assumption that the 3D cell culture is relatively closer to in vivo than does the 2D does the percent toxicity caused by the highest concentration of your compounds could be a concern in an animal system. Do you see that as a potential problem? Please, offer your comment/response.
Authors: We do not consider that it would be a potential problem, since the NPs toxicity was evidenced in the resistant tumor cells and the formulation showed low cytotoxicity in non-tumor cells, which suggests that the NPs are selective to the tumor cells. Therefore, we expect that the co-loaded NP would also achieve higher antitumor effects in an animal model than the association of the free compounds. However, even though 3D in vitro models are quite suitable for predicting in vivo effects, the higher complexity of the animal models can result in different effects of NPs in vivo. Due to the promising results obtained here using in vitro conditions, as a second part of this study, we aim to evaluate the in vivo performance of the AFAT-Se-PTX-PCL-77KS-NP.
- Your results support the possibility your compounds will be safe for cancer treatment. But did you determine its potential effects on immune system by examining their effects on T-cells viability/T cell activation?
Authors: We understand the reviewer’s concern. However, the main objectives of this study and the resources available for its conduction were focused on evaluating the antitumor effectiveness of the innovative combination of bioactive compounds intended for an anticancer treatment. Despite acknowledging the crucial importance of the immune system in this context, the decision not to include it was due to the priorities outlined and the challenging experimental implementation.
- Did you determine the effects of the conjugated compounds on cytokine production? This crucial because your compounds could induce expression and secretion of pro-inflammatory cytokine expression (IL-1beta, IL-6 and TNF), which could alter the tumor microenvironment and hence be detrimental to cancer patients who may be treated with these NPs.
Authors: We appreciate the consideration and recognize the significance of assessing pro-inflammatory cytokines within the scope of our research. However, our laboratory currently lacks the necessary materials to conduct such analyses. It's important to note that the NPs under investigation represent a novel contribution to the field. As we pursue further studies aimed at exploring synergistic associations within NPs, collaborations may offer avenues for elucidating these aspects.
- If you did not determine the effects of your NPs on cytokine expression in cancer cell lines, it may be better to mention them in your discussions as potential limitations of your studies.
Authors: As recommend, we included the limitations of our study in the end of the discussion section. The text was adjusted as follows:
Line 665: “The overall results of this study are promising, but it is indeed important to recognize the limitations for future advancements, given the significance of the innovative combination proposed here. The employed cellular models are well-recognized and commonly used; however, they may not fully capture the in vivo context and complexity, thus limiting the generalization and immediate correlation of the results, both in terms of activity and toxicity in a clinical context. Additional studies, including animal models, might be the next steps to take. Long-term effects of the nanoformulation could also be of interest, including assessments involving its role on platelets function, as well as studies on its impact on the immune system and cytokine expression in the tumor microenvironment.”
- Minor concerns:
There are a few grammatical errors that can easily be corrected.
Examples: On line 525, the word determine should determined
Line 605, corroborate should corroborated
Line 618, growth should be growth
Authors: We thank the reviewer for the valuable corrections. The text was adjusted, as recommended.
Line 554: “The encapsulation efficiency was successfully determined by ….”
Line 640: “Therefore, the observed synergism may also be corroborated ….”
Line 653: “….which have a considerable growth.”

Round 2
Reviewer 1 Report
Comments and Suggestions for Authors
The authors have addressed the questions raised with satisfactory responses. The paper can be accepted in the current format
Comments on the Quality of English LanguageThe authors have addressed the questions raised with satisfactory responses. The paper can be accepted in the current format
Reviewer 2 Report
Comments and Suggestions for Authors
The manuscript has been improved. The authors have addressed the reuqested revisions. Now the manuscript is suitable for publication.
Reviewer 3 Report
Comments and Suggestions for Authors
Overall, their original manuscript was excellent. However, there were a few vital concerns identified by the reviewer. Those concerns made it impossible to recommend an acceptance of the manuscript at that time. However, in the revised version, the authors have positively addressed those concerns. In addition, the authors have incorporated their responses into the appropriate areas of the manuscript. The authors have also incorporated additional references to further strengthen the overall quality of the manuscript.